

# Chemical and physical characterization of traffic particles in four different highway environments in the Helsinki metropolitan area

J. Enroth[1,2], S. Saarikoski[3], J. V. Niemi[4,5], A. Kousa[4], I. Ježek[6], G. Močnik[6,7], S. Carbone[3,9], H. Kuuluvainen[8], T. Rönkkö[8], R. Hillamo[3], and L. Pirjola[1,2]

[1]Department of Technology, Metropolia University of Applied Sciences, P.O. Box 4021, FI-00180 Helsinki, Finland
[2]Department of Physics, University of Helsinki, P.O. Box 64, 00014 Helsinki, Finland
[3]Atmospheric Composition Research, Finnish Meteorological Institute, P.O. Box 503, 00101 Helsinki, Finland
[4]Helsinki Region Environmental Services Authority HSY, P.O. Box 100, FI-00066 HSY, Helsinki, Finland
[5]Department of Environmental Sciences, University of Helsinki, Helsinki, Finland
[6]Aerosol d.o.o., Ljubljana, Slovenia
[7]Condensed Matter Physics Department, Jožef Stefan Institute, Ljubljana, Slovenia
[8]Department of Physics, Tampere University of Technology, P.O. Box 692, 33101 Tampere, Finland
[9]now at: Department of Applied Physics, University of São Paulo, São Paulo, Brazil

*Correspondence to*: L. Pirjola (liisa.pirjola@metropolia.fi, liisa.pirjola@helsinki.fi)

**Abstract.** Traffic related pollution is a major concern in urban areas due to its deleterious effects on human health. The characteristics of the traffic emissions on four highway environments in the Helsinki metropolitan area were measured with a mobile laboratory, equipped with state-of-the-art instrumentation. Concentration gradients were observed for all traffic related pollutants, particle number (CN), particulate mass $PM_{2.5}$, black carbon (BC), organics and nitrogen oxides (NO and $NO_2$). Flow dynamics in different environments appeared to be an important factor for the dilution of the pollutants. For example, the half-decay distances for the traffic-related CN concentrations varied from 8 m to 83 m at different sites. The $PM_1$ emissions from traffic mostly consisted of organics and BC. At the most open site, the ratio of organics to BC increased with the distance to the highway indicating condensation of volatile and semi-volatile organics on BC particles. This condensed organics was shown to be hydrocarbons as the fraction of hydrocarbon fragments in organics increased. Regarding the CN size distributions, particle growth during the dilution was not observed, however, the mass size distributions measured with a soot particle aerosol mass spectrometer (SP-AMS), showed a visible shift of the mode, detected at ~100 nm at the roadside, to a larger size when the distance to the roadside increased. The fleet average emission factors for the CN appeared to be lower and for the $NO_2$ higher than ten years ago. The reason is likely the increased fraction of light duty (LD) diesel vehicles in the past ten years. The fraction of heavy duty traffic, although constituting less than 10 % of the total traffic flow, was found to have a large impact on the emissions.

## 1 Introduction

Vehicle exhaust emissions constitute major sources of ultrafine particles (UFP, below 100 nm in diameter), black carbon (BC), organic carbon (OC), and $NO_2$ in urban environments (e.g. Pey et al., 2009; Morawska et al., 2008; Johansson et al., 2007;





Lähde et al., 2012). Although during the last 15 years particle mass emissions have been significantly reduced due to the tightened emission regulations and improvements in vehicle technology, the number emissions of the smallest UFP particles (below 50 nm in diameter) have been observed to be significant (Rönkkö et al., 2013; Kumar et al., 2011). Besides the exhaust emissions participate in chemical and physical transformation processes in the atmosphere affecting urban visibility and global

climate (IPCC, 2013), they have harmful health effects. Ultrafine particles can penetrate deep into the human pulmonary and blood-vascular systems increasing the risk to get asthma, reduced lung function, cardiovascular disease, heart stroke, and cancer (Pope III and Dockery, 2006; Sioutas et al., 2005; Kettunen et al., 2007; Su et al., 2008; Alföldy et al., 2009). The European Environment Agency has estimated that fine particles caused around 430 000 premature deaths in Europe and around 1900 in Finland in year 2012 (EEA, 2015). Particularly, people who live, work or attend school near major roads have an

increased health risk.

During the last decade, pollutant gradients near highways have been extensively examined in USA (Zhu et al., 2009; Clements et al., 2009; Hagler et al., 2009; Durant et al., 2010; Padró-Martínez et al., 2012; Massoli et al., 2012), in Canada (Beckerman et al., 2008; Gilbert et al., 2007), in Australia (Gramotnev and Ristovski, 2004), in India (Sharma et al., 2009), and in Finland (Pirjola et al., 2006). Generally, all of these studies showed that the pollutant concentrations were higher near

highway than further from the roadside, and that they sharply decreased to background levels in 300-500 m distances from the roadside downwind. However, Gilbert et al. (2007) discovered that the $NO_2$ concentration decreased during the first 200 m distance from the edge of the highway but beyond 200 m downwind started to increase indicating that other factors than the highway traffic influenced the increased $NO_2$ concentration.

The studies have shown that the concentration levels and gradient shapes of UFP and other primary vehicular emissions

near major roads depend on many factors, including meteorological conditions such as atmospheric stability, temperature, wind speed, wind direction, and surface boundary layer height (Durant et al., 2010). Traffic fleet and flow rate (e.g. Zhu et al., 2009; Beckerman et al., 2008), background concentrations (Hagler et al., 2009), and atmospheric chemical and physical processes (Beckerman et al., 2008; Clements et al., 2009), all affect pollutant concentrations near the highways. Hagler et al. (2009) and Janhäll et al. (2015) found that local topography and land use, particularly noise barriers and roadside vegetation,

can also be important factors determining the concentrations. In addition, the measurement results depend on sampling techniques and instruments used in studies.

Using single particle mass spectrometry the characteristics of vehicle emissions have been recently studied in dynamometers (e.g. Sodeman et al. 2005; Shields et al., 2007) and near highways (e.g. Lee et al., 2015). Only a few of the published studies have investigated changes in exhaust particle chemical composition during dispersion. Clements et al. (2009)

collected high-volume $PM_{2.5}$ samples at 35 and 65 m distances from a major highway. They discovered that unlike the particle-bound elemental carbon (EC), organic carbon (OC) concentrations increased with the distance downwind. Instead, Durant et al. (2010) report a time-dependent decrease in the concentrations of particulate organics (Org) and hydrocarbon-like organic aerosol (HOA) up to 200-300 m downwind from a highway during morning hours. Massoli et al. (2012) present spatial-temporal gradients of the HOA and oxygenated organic aerosol (OOA) concentrations summed with the refractory BC (rBC)





up to 500 m downwind from a highway. The sum of HOA and rBC mass concentration decreased with increasing distance whereas the sum of OOA and rBC was constant. The size distributions of organics and rBC pointed out that the fresh soot mode peaking at ~100 nm was slightly coated by the HOA material whereas the accumulation soot mode peaking at ~500 nm was heavily coated by the OOA material representing the background aerosol. The change in the chemical composition of traffic particles with the distance is caused by several reasons. The exhaust from the vehicles is hot when emitted but it cools quickly as it is mixed with ambient air. Cooling promotes the condensation of organic vapours on particles but as the exhaust is diluted with ambient air, concentration of gaseous semivolatile organic compounds (SVOCs) is reduced, leading to the evaporation of SVOCs from particles in order to maintain phase equilibrium (Robinson et al., 2010). So there is an ongoing competition between different processes in the emission plume; new particle formation (nucleation), particle growth through condensation and coagulation, and decrease of particle mass by evaporation.

The objective of this study was to characterize the spatial variation of traffic-related air pollutants downwind from the four highways in the Helsinki metropolitan area during rush hours. The measurements were performed by a mobile laboratory van "Sniffer", equipped with high time-resolution instrumentation. In addition to gaseous pollutants and particle number (CN), mass (PM) and size distribution, also the chemical composition of particles was measured. This study addresses the following questions: 1) how different environments affect dilution and concentration gradients, 2) how the properties of CN size distribution change as a function of distance and location, 3) how the chemical composition of the particles evolves as the pollutants move away from the road, and 4) what are the emission factors of the main pollutants in different environments. For the first time, pollutant gradients near several highways were investigated from various aspects by combining the physical measurements with the detailed chemical speciation by using the state-of-the art-instrumentation in a mobile measurement platform.

## 2 Experimental methods

### 2.1 Measurement sites and sampling strategy

Measurements of gaseous and particle pollutants were conducted at four different environments (Fig. 1) located next to three highways in the Helsinki metropolitan region. The highways Ring I and Ring III have two traffic lanes in each direction whereas the highway Itäväylä has three lanes in each direction. Each of the four measurement sites had a small dead end road with minimal traffic perpendicular to the highway. Measurements were performed by a mobile laboratory van Sniffer at four different locations: (1) driving within the traffic at the highway, (2) stationary measurements next to the highway, (3) approaching the highway by driving with an average speed of $6.6 \pm 1.5$ km/h along the small road (gradient measurements) and (4) background concentrations were measured for each environment at a suitable remote location approximately 500 m away from the highway. The gradient measurement periods and sites were selected so that wind was blowing from the direction of the highway to the measurement road. Because the concentration field varied spatially and temporally, the gradient measurements were performed up to eleven times per measurement occasion on each road. Depending on the location and the



length of the measurement road, a single approach took about two minutes. Some of the approaches suffered from cars passing close by, sudden gusts of wind or other disturbances, and these were excluded. A total of 89 successful approaches were recorded. The approaches were carried out so that sampling inlets were on upwind side in order to avoid Sniffer's own exhaust. Measurements were performed during rush hours, 7-10 am and 3-6 pm, on the period from 22 October to 6 November 2012.

During the stationary measurements by the highway the drivers manually calculated traffic flow for heavy duty (HD: trucks and buses) and light duty (LD: passenger cars and vans) vehicles during three minutes to each direction. The traffic counts, taken at each site, are summarized in Table 1.

For the northern wind, the gradient measurements were performed along Isovaarintie in Espoo, next to Ring III near Lake Pitkäjärvi (Fig. 1). The environment was very open, surrounded only by the fields. The maximum distance to Ring III from

10 Isovaarintie was 250 m. The gradients were measured while approaching the Ring III from the end of Isovaarintie. The roadside measurements were taken from a stationary position 14 m from the Ring III. According to Finnish Transport Agency (2015) the annual mean traffic flow on the Ring III was around 40 000 vehicles per day, 4 000 of which were HD vehicles. The percentage of HD vehicles on this site, based on our manual three minute observations, was 8.7%. This was also the highest share of HD vehicles of the four investigated sites (Table 1). The speed limit on Ring III was 100 km/h.

For the southern wind, the gradient measurements were performed at Malmi from 11 m to a maximum of about 260 m from Ring I (Fig. 1). Malmi is a suburban semi-open area surrounded by few buildings and trees. There is another fairly busy road at the far end of the measurement road. The manual calculations showed that 4.8% of vehicles were HD vehicles (Table 1), whereas the annual mean traffic flow of Ring I at Malmi is around 55 000 vehicles per day of which about 3 300 are HD vehicles (Finnish Transport Agency, 2015). The speed limit on Ring I was 80 km/h. The Helsinki Region Environmental

Services Authority (HSY) had an air quality monitoring station at Malmi, and the chemical composition and sources of $PM_1$ at this site have been discussed earlier in Aurela et al. (2015).

In the case of the southern wind, the measurements were also carried out at Itä-Pakila on Klaukkalantie next to Ring I (Fig. 1). Itä-Pakila is a fairly uniform area with tightly built small houses with gardens (allotment). The highway is separated from the residential area with a noise barrier. Mostly it is a land barrier with a row of spruce trees planted on its top, the height being

around 5 m. Only at the westernmost part (~ 100 m from the gradient start) the barrier is around 2 m high wooden fence. At the location of the gradient measurements the wooden fence has an opening for pedestrian traffic, and a pedestrian bridge crossing the highway (Fig. 1). In addition to the roadside and gradient measurements, the pedestrian walkway behind the noise barrier was also measured on each of the routes. At Itä-Pakila the maximum distance of the gradient measurements was about 260 m from the highway. The roadside measurements were performed on a bus stop next to Ring I (6 m from the roadside).

The manual count of mean traffic flow on Ring I next to Itä-Pakila was highest of the four investigated sites, about 5 900 LD and 160 HD vehicles per hour. It was the highest also according to the authorities, which report 57 000 vehicles per day with of which 3 200 are HD vehicles (Finnish Transport Agency, 2015). The speed was limited to 80 km/h.

For the northwesterly wind, the gradient road used for highway Itäväylä was at Herttoniemi, Helsinki (Fig. 1). The maximum distance to Itäväylä was 130 m. The Herttoniemi measurement site is in a semi-industrial area, with a rough surface





environment. At Herttoniemi most measurements were performed as stationary measurements at the distances of 11 (roadside), 20, 30, 40, 50, 70, 100 and 130 m from Itäväylä. The mean traffic flow on Itäväylä is around 50 000 vehicles per day (Finnish Transport Agency, 2015), and the speed limit was 80 km/h. The Herttoniemi site was partially chosen because it had already been used nine years ago in the LIPIKA project (Pirjola et al., 2006), thus enabling the comparison of the results.

During the campaign, the weather was rather mild, with the temperature around 0.8-4.7ºC, relative humidity 77-89%, and wind speed around 3-5 m s$^{-1}$, monitored at the meteorological measurement site at Ämmässuo (Fig. 1) by the HSY, representing the regional air mass properties. A summary of the meteorological and traffic conditions at each site is presented in Table 1.

### 2.2 Instrumentation

Measurements were performed with a mobile laboratory van "Sniffer" (VW LT35 diesel van) described in detail in Pirjola et

al. (2004, 2006, 2012, 2015). The inlets were positioned above the van's windshield, 2.4 m above the ground level. During the stationary measurements, the engine was switched off and the data of the first three minutes was excluded. The list of the instruments is given in Table S1 in the Supplement and only shortly desribed below.

Particle number concentrations and size distributions were measured with two ELPIs, (Electrical Low Pressure Impactor, Dekati Ltd.) (Keskinen et al., 1992) both equipped with a filter stage (Marjamäki et al., 2002). Furthermore, an additional stage

designed to enhance the particle size resolution for nanoparticles was installed into one ELPI (Yli-Ojanperä et al., 2010). The particle size distribution was also measured with an EEPS (Engine Exhaust Particle Sizer, model 3090, TSI). The measurement ranges of the ELPIs and EEPS were 7 nm - 10 μm and 5.6 - 560 nm, respectively, and the time resolution of one second was fast enough to register dynamic changes in the traffic exhaust while driving. It should be noted that the ELPIs measure particle aerodynamic diameters while the EEPS measures particle electrical mobility diameter. The number concentration of particles

larger than 2.5 nm was measured by a butanol CPC (3776, TSI) with a time resolution of one second.

To study particle volatility characteristics, a thermodenuder (TD; Rönkkö et al., 2011) was installed in front of the ELPI which did not have an additional stage. In the TD, the diluted sample was heated to 265 °C and after that, led into the denuder where the cooled inner wall was covered with activated carbon to collect evaporated compounds. The particle size distributions measured after the TD were corrected for particle losses (Heikkilä et al., 2009).

Black carbon (BC) in the PM$_1$ size fraction (using a cyclone) was measured with an Aethalometer (Magee Scientific Model AE33) with a one second time resolution. Measurements at 880 nm were used for the reported BC concentrations. The data was compensated for the loading effect using Drinovec et al. (2015), compensation algorithm with ten seconds time resolution.

A DustTrak (TSI, model 8530) with a 2.5 μm cut-off was used to measure the real-time PM$_{2.5}$ concentration with a time resolution of one second. The DustTrak operates using a light scattering technique where the amount of the scattered light is

proportional to the volume concentration of the aerosol. Since the instrument was factory calibrated with Arizona test dust particles, the readings generally need to be corrected for the aerosol type to be measured. Based on our previous study (Pirjola et al., 2012), all DustTrak PM$_{2.5}$ data were divided by a factor of 1.46 for a particle density correction.



For this study the Sniffer was also equipped with a SP-AMS (Soot Particle Aerosol Mass Spectrometer, Onasch et al., 2012) to study particle chemistry. In the SP-AMS, an intracavity Nd:YAG laser vaporizer (1064 nm) is added into the High Resolution Time-of-Flight Aerosol Mass Spectrometer (HR-ToF-AMS, DeCarlo et al., 2006) in order to measure rBC and associated non-refractory particulate material (e.g. metals) in addition to the non-refractory species, sulfate ($SO_4$), nitrate

($NO_3$), ammonium ($NH_4$), chloride (Chl) and organics (Org). In this study the SP-AMS measured in mass spectrum (MS) mode with five seconds time resolution of which half of the time the chopper was open and half of the time closed. In addition to MS mode, unit mass resolution (UMR) particle Time-of-Flight (pToF) data was collected at Pitkäjärvi and Herttoniemi in order to obtain the mass size distributions for the chemical species. There was a $PM_1$ cyclone in front of the SP-AMS but the real measured size range of the instrument is ~50–800 nm (Canagaratna et al., 2007). The SP-AMS data was analyzed using

a standard AMS data analysis software (SQUIRREL v1.57 and PIKA v1.16) within Igor Pro 6 (Wavemetrics, Lake Oswego, OR) and for the elemental analysis of organics an Improved-Ambient method was used (Canagaratna et al., 2015). The mass concentrations from the SP-AMS data were calculated by using a collection efficiency of 0.5 (Canagaratna et al., 2007 and references therein). Even though both the aethalometer and SP-AMS can measure BC, only the data from the aethalometer is used in this paper for the concentrations of BC. The reason for this was that the SP-AMS gave much (~70%) smaller

concentrations for rBC than aethalometer for BC likely to due to the nonoptimal laser-to-particle beam vertical alignment previously discussed e.g. in Massoli et al. (2012). However, for the mass size distributions (Section 3.3.2) rBC from the SP-AMS is used as the aethalometer cannot give BC concentration as a function of the particle size. The SP-AMS has been used previously in traffic related measurements in e.g. Massoli et al. (2012), Dallmann et al. (2014) and Pirjola et al. (2015).

Gaseous concentrations of $CO_2$ (model VA 3100, Horiba), CO (model CO12M, Environnement S.A.), and nitrogen oxides

NO, $NO_2$ and $NO_X$ (model APNA 360, Horiba) were monitored with a time resolution of one second. A weather station (model WAS425AH and model HMP45A, Vaisala) on the roof of the van at a height of 2.9 m above the ground level provided meteorological parameters. Additionally, a global positioning system (model GPS V, Garmin) recorded the van speed and position.

## 2.3 Data handling

For each site, the data was averaged as a function of distance from the curb of the road. The average values were calculated at 25 m intervals from 25 m up to 300 m from the curb, with the ending point depending on the location. Each distance $i$ on the gradient represents the span from $i$-12.5 m to $i$+12.5 m. The distances were determined from the GPS data.

## 2.4 Emission factor calculations

Fleet emission factors were calculated for various pollutants using a method adapted from Yli-Tuomi et al. (2004). The fuel

based emission factor indicates how much of a given pollutant is emitted per amount of fuel consumed. Since the modern car fleet has very high combustion efficiency, we can assume that all carbon in the fuel was converted into $CO_2$. The emission factor for pollutant X can be expressed as



$$EF_X = \frac{CMF_{CO_2}(X - X_{bg})}{CMF_{fuel}(CO_2 - CO_{2,bg})} \tag{1}$$

where $X$ is the pollutant concentration, and $X_{bg}$ and $CO_{2,bg}$ are the background concentrations. For the $CO_2$ production rate, we adopted a value provided by the Technical Research Centre of Finland, $CMF_{CO2}/CMF_{fuel}$ = 3,141 g (kg fuel)$^{-1}$, the same as used by Yli-Tuomi et al. (2004). Similar values were also reported in other papers like Ježek et al. (2015). In order to characterize the fleet more representatively, two minute averages were used in the emission factor calculations.

## 3 Results and discussion

### 3.1 Overview of the concentration gradients

Rapidly decreasing concentrations from the highway were observed for particle number and mass as well as gases on all four investigated locations. For each site, Table 2 summarizes the average concentrations over the entire measurement time while driving on a highway, while being parked at the roadside and at the background locations, along with the respective standard deviations. As in previous studies (Zhu et al., 2002, 2009; Pirjola et al., 2006; Massoli et al, 2012), the CN were found to drop rapidly and level out to slightly above background levels from 100 to 300 meters from the roadside. For example, the average CN on the highways varied from $(7.7\pm9.1)\times10^4$ cm$^{-3}$ at Herttoniemi to $(12.2\pm14.0)\times10^4$ cm$^{-3}$ at Pitkäjärvi, and the average background concentrations from $(7.1\pm2.0)\times10^3$ cm$^{-3}$ at Herttoniemi to $(10.9\pm2.0)\times10^3$ cm$^{-3}$ at Malmi. Considerable differences between the sites were found in the particle dilution. Figure 2 illustrates the normalized curves for the behavior of CN, PM$_{2.5}$, BC, organics, NO and NO$_2$, for each site when the background concentrations were first subtracted and then the concentrations were divided by the concentrations measured at the highway. Most rapid decrease was observed at Itä-Pakila where a 50% reduction in the CN already occurred at a distance of 8 m from the highway (Table S2). The exceptional dilution of pollutants at this site was obviously caused by the noise barrier, since the gradient route went through a narrow gap between the barrier ends. However, the measurements on the pedestrian walkway behind the noise barrier showed a large variation in the CN concentration depending on the height and type of the noise barrier (Fig. S1). The half-decay distance at Malmi was 83 m, and around 40 m at Herttoniemi and Pitkäjärvi, based on the fitted curves in Fig. 2 (Table S2). According to the earlier studies (Pirjola et al., 2006; Fig. 9), the average CN concentration downwind Itäväylä at Herttoniemi was reduced to half of the concentration at the roadside at ~ 55 m from the middle of the highway, i.e. ~ 40 m from the roadside.

Highest PM$_{2.5}$ concentrations measured at the highway were detected at Itä-Pakila (19.5 ± 11.6 µg m$^{-3}$) followed by Malmi (16.0 ± 22.5 µg m$^{-3}$), and the lowest highway PM$_{2.5}$ was found at Pitkäjärvi (10.1 ± 8.9 µg m$^{-3}$) (Table 2). The highest roadside PM$_{2.5}$ concentration was as well measured at Itä-Pakila (15.5 ± 4.6 µg m$^{-3}$), and the lowest was found at Herttoniemi (7.3 ± 4.0 µg m$^{-3}$) where the background concentration was lowest as well. Rather similar reductions than for the CN can be observed for the PM$_{2.5}$ concentrations. The strongest dilution occurred at Itä-Pakila, the concentrations reduced to half of the highway concentration at 13 m at Itä-Pakila, and 33-41 m at the other sites (Table S2).


The terrain dynamics of the different measurement sites appeared to be an important factor for pollution dilution. The open environment of Pitkäjärvi produced smooth pollution gradients on most runs while the more urbanized environments of Herttoniemi and Malmi had considerably more variation present in their gradients. At Malmi, the presence of the second road at the end of the measurement lane (at the distance of 175-200 m from Ring I) was also apparent in the data, often resulting in

a U-shaped pollution profile. The noise barrier lowered the pollutant concentrations at Itä-Pakila considerably. Previous studies have found that sound barriers can create constant local eddies in the wake of the sound barrier, resulting in a lower pollution zones (Boweker et al., 2007, Ning et al., 2010).

For the gaseous pollutants, the dilution rates were similar to those of the particle concentrations, i.e. rapid decrease in the first tens of meters and levelling out at 100-300 meters to the urban ambient levels (Table 2). The exceptionally rapid dilution

at Itä-Pakila was nod also with the NO concentrations (Fig. 2). Consequently, the ratio of $NO_2$ to NO was higher than unity already at 50 m distance from the roadside whereas at least 100 m was needed at the other sites (Table S3).

In a similar study by Massoli et al. (2012), they did not observe a gradient for $NO_2$, but found it to be dependent on the time of day. Thus they linked it to photochemical conversion, and concluded that $NO_2$ is not an efficient indicator of traffic pollutants on a short time scale. We observed rather similar behavior for $NO_2$ at Malmi, whereas a clear spatial $NO_2$ gradient

near the highways could be observed at Itä-Pakila, Pitkäjärvi, and Herttoniemi. The observed gradient of $NO_2$ is likely explained by the difference in the vehicle fleet composition and the background $NO_2$ levels between New York and Helsinki. In Helsinki, the fraction of light duty diesel vehicles of the passenger cars is very high, 34.3 % (Official statistics of Finland, 2015), and thus there seems to be sufficient amount of $NO_2$ directly emitted from traffic to form an observable gradient. The sunrise and sunset during the measurement period coincided with the rush hours, thus making the analysis of photochemistry

more difficult. NO did show time dependent behavior with higher concentrations present in the morning rush hour (Fig. S2). In the afternoon, it seems as though the emitted NO rapidly converted into $NO_2$ by $O_3$ oxidation, and as a result lower levels of NO were observed.

## 3.2 Particle number size distributions

Figure 3 shows the average number size distributions over all measured periods at different environments, recorded by the

EEPS. Three modes can be observed, the nucleation mode peaked at ~10 nm, Aitken mode at ~30-40 nm and the soot mode at ~70-80 nm. Sometimes, the nucleation mode had two peaks, one at ~10 nm and the other at 16-20 nm. The exact shape of the size distribution was observed to be dependent on the location. For example, the Aitken mode was largest at Itä-Pakila where the traffic flow was highest, consisting mostly of LD vehicles. Instead, the nucleation mode was highest at Pitkäjärvi where the traffic was not that busy but consisted more of HD vehicles.

Pirjola et al. (2006) observed particle growth in the nucleation mode with increasing distance from the highway during a previous winter campaign. Here, we did not observe significant growth of the mean diameter of the nucleation mode particles. When considering the whole size distribution in the size range of 5.6-560 nm, the average diameter (Table S4) grew by 1.7%, 2.1%, 22% and 17% at 100 m from the road for Herttoniemi, Malmi, Itä-Pakila and Pitkäjärvi, respectively, compared to the





average diameter observed on the highway. It is plausible that this growth mostly resulted from mixing with the background particles, that on average were larger than the freshly emitted particles at all sites except at Herttoniemi (Table S4). During dispersion the smallest particles decreased faster than the larger ones. For example, at 100 m distance the particles in the size ranges from 6-30 nm, 30-60 nm, 60-150 nm and 150-500 nm had decreased in their respective concentrations by 76, 68, 64

and 60% compared to the concentrations measured on the highways.

Particle volatility was studied by two ELPIs, one before and the other after the TD treatment. Figure 4, presenting the results from Herttoniemi, indicate that particle volatility was size dependent. The smallest particles (< 30 nm) were found to be highly volatile indicating that the origin for these particles might be nucleation of sulphuric acid from fuel and lubricant oil sulphur compounds along with volatile organic compounds (Arnold et al., 2012, Kittelson et al., 2008). The existence of non-

volatile cores (e.g. Rönkkö et al., 2007) in sub 30 nm particles could not be estimated mainly because particles smaller than 7 nm cannot be  measured by the ELPIs.

The soot mode concentrations showed lowest reductions after the TD treatment. The size distribution after the TD treatment peaked at around 70 nm by number and at ~200 nm by volume (aerodynamic diameter) which coincides with the typical size of soot particles from traffic emissions. The TD treatment reduced particle number and volume in 7-1000 nm size range by

86% and 65% respectively, showing that most of the particle material was volatilized at high temperatures.

**3.3 Chemical composition of traffic particles**

Particles at the highway and roadside comprised mostly of BC and organics (Table 2). The contribution of BC to $PM_1$ (sum of chemical species measured with the SP-AMS and aethalometer) was 54, 40, 28 and 41% at the highway at Herttoniemi, Malmi, Itä-Pakila and Pitkäjärvi, respectively, with the corresponding contribution of organics being 41, 46, 54 and 51%. At the

background locations, the particles were mainly made of organics and sulfate (50 and 42% at Malmi, 52 and 35% at Itä-Pakila and 44% and 43% at Pitkäjärvi, respectively) or organics and BC (60 and 26% at Herttoniemi). At Malmi and Itä-Pakila there were also some nitrate and ammonium in the particles at the background location (11 and 8% at Malmi and 13 and 9% at Itä-Pakila, respectively), and at Malmi the particles had a minor fraction of chloride (3%). In addition to BC, organics and inorganic salts, particles were found to contain trace amounts of metals. Metals will be discussed separately in Section 3.3.3.

An example of the chemical composition of $PM_1$ particles measured at different distances from the road is presented in Fig. S3 at Pitkäjärvi. Only the major components are included in the figure and therefore, e.g. chloride and the metals, are not shown. As seen from Fig. S3, the mass fraction of BC decreased with the increasing distance from the road, whereas the fractions of organics and inorganics (sulfate, ammonium and nitrate) increased the contribution of background aerosol becoming more predominant as moved further from the road.

Normalized dilution curves of organics and BC are presented in Fig. 2. For both organics and BC the concentrations decreased fastest at Itä-Pakila noise barrier site where these concentrations dropped to half of those at the highway already at the roadside and nine meters from the road, respectively (Table S2). Except at Itä-Pakila, the dilution curves for organics were quite similar at all other sites. Regarding BC, the dilution curves had similar trends at Herttoniemi and Pitkäjärvi, whereas at





Malmi the concentration decreased up to 100 meters from the road after which it remained at elevated level for the rest of the gradient. In general, organics reached 50% reduction much earlier than BC except at Pitkäjärvi. The average half-decay values over all sites for organics and BC were ~24 and 33 meters, respectively (Table S2).

Figure 5a shows that the ratio of organics to BC varied from 0.58 to 1.34 at the highway. The ratio was smallest at Herttoniemi and largest at Itä-Pakila. When moving away from the highway, the evolution of the ratio was rather different at different sites. The ratio varied significantly at Itä-Pakila, and therefore it is shown only with separate points in Fig. 5a. High variation was probably due to the rapid decrease of traffic pollutants at Itä-Pakila (because of noise barrier), and thus a high uncertainty in the calculation of organics to BC ratio as the ratio was calculated only for traffic related particles after background subtraction. At Malmi the ratio of organics to BC was clearly larger at 25 meters from the road whereas it was rather stable at all other measurement points. At Herttoniemi and Pitkäjärvi the ratio of organics to BC increased with the distance. At Herttoniemi the ratio increased slightly from the roadside up to 100 meters, whereas at Pitkäjärvi (open field site) the ratio increased, however not very smoothly, all the way from the roadside to the last measured gradient distance, 250 meters from the road. At 250 meters from the road the ratio of organics to BC was already double what it was at the highway at Pitkäjärvi. The increase of this ratio is assumed to be associated with the condensation of volatile and semi-volatile organics on BC particles when hot exhaust aerosol was mixed with ambient air and cooled. In general, during the dilution of exhaust aerosol, there is a competition between particle formation, particle growth via condensation and coagulation, and reduction of particle mass with evaporation. Chemical composition of organics and its size distributions in traffic particles will be investigated in detail in next two sections.

For nitrate, sulfate and ammonium, no change in the concentrations with the distance was observed (Fig. S4). This is expected as vehicles are not significant direct emitters of particulate nitrate, and the use of ultralow sulfur diesel fuel results in very low emissions of particulate sulfate. Similar results for sulfate, ammonium and nitrate have been shown e.g. in Durant et al. (2010). However, at Herttoniemi and Pitkäjärvi the chloride concentrations were slightly larger at the highway, roadside and near the road (≤50 m distance) than at the other measurement points (Fig S4d). Chloride concentrations could be related to the lubricating oil, or in a small part to the road salt used in Finland in wintertime.

### 3.3.1 Traffic-related organics

The composition of organic matter was investigated by dividing organic fragments based on their elemental composition. In addition to carbon and hydrogen atoms, organic matter consists of oxygen and, typically in small amounts, nitrogen and sulfur atoms. The chemical composition of organics at the highway, roadside and background at all four sites are shown in Fig. S5. Similar to the previous studies (e.g. Canagaratna et al., 2004; Chirico et al., 2011) most of the organics consist of hydrocarbon fragments $(C_XH_Y{}^+)$ at all measurement locations. The fraction of hydrocarbons decreased from highway to background, the portion of hydrocarbons being on average 77, 70 and 53% at the highway, roadside and background, respectively. The largest single fragments in $C_XH_Y{}^+$ group were $C_4H_9{}^+$ (at $m/z$ 57), $C_3H_7{}^+$ (at $m/z$ 43), $C_4H_7{}^+$ (at $m/z$ 55) and $C_3H_5{}^+$ (at $m/z$ 41; Fig. S5). Most of the other organics were made of oxygen-containing fragments. Organic fragments with one oxygen atom $(C_XH_YO^+)$



had slightly larger fraction in organics than fragments with more than one oxygen atom $(C_XH_YO_Z^+, {}_{Z>1})$, especially at the background sites. $C_XH_YO^+$ group had largest signal for $C_2H_3O^+$ (at *m/z 43*), $CO^+$ (at *m/z 28*) and $CHO^+$ (at *m/z 29*), fragments whereas $C_XH_YO_Z^+, {}_{Z>1}$ group consisted almost entirely from $CO_2^+$ (at *m/z 44*). There was also an indication of nitrogen-containing organics $(C_XH_YN^+)$ in traffic particles, however, they were difficult to separate from neighboring peaks in the MS as they constituted less than 1% of all organics.

When comparing the sites, the fractions of hydrocarbons and oxidized organics were similar at all highway sites whereas the fractions of oxidized organics at the roadside and background were larger at Itä-Pakila than at Herttoniemi, Malmi and Pitkäjärvi. This was likely due to the high contribution of long-range transport because elevated levels of nitrate, sulfate and ammonium were observed at all the measurement positions at Itä-Pakila site (Fig. S4). In line with the higher portion of oxygenated organic fragments, the ratios of oxygen to carbon (O:C) and organic matter to organic carbon (OM:OC) at the highway, roadside and background were larger at Itä-Pakila than at any other site (Fig. S5).

The fractions of hydrocarbons were smaller in this study than that in Dallman et al. (2014) measured in the San Francisco Bay area. They found that the family $C_XH_Y^+$ contributed 91% of the measured organics signal whereas the families $C_XH_YO^+$ and $C_XH_YO_Z^+, {}_{Z>1}$ contributed less than 10%. However, they measured the vehicle emissions in a highway tunnel where the contribution of background organics was assumed to be smaller than in this study.

Concentration gradients for hydrocarbon and oxygen-containing organic fragments after the background subtraction are shown in Fig. 6a. It is clear that hydrocarbon concentrations decrease with the distance from the road but for oxidized fragments the concentration depended less on the distance. At Herttoniemi, both oxidized fragments ($C_XH_YO^+$ and $C_XH_YO_Z^+, {}_{Z>1}$) clearly fell off with the distance from the road, whereas at all the other sites the concentrations of the oxidized fragments were typically slightly larger at the highway. However, a decreasing trend from the road was not observed. The lack of any significant spatial gradient for oxidized fragments suggests that they correspond mostly to the aged background aerosol. Similar to this study, Canagaratna et al. (2010) observed a concentration gradient for hydrocarbon-like organic aerosol in Massachusetts, USA. Regarding the oxygenated organic aerosol they found an increase up to 150 meters from the road after which the OOA concentration decreased. It should be noted in Fig. 6 that some of the concentrations for the oxidized fragments are negative. That indicates that the measured concentrations were smaller than those measured at the background.

Organics in the engine exhaust particles originate from unburned fuel and lubricant oil as well as their partially oxidized products. Different processing technique for fuel and lubricant oil leads to large differences in their molecular weights and chemical structures. This results in divergent mass spectra (MS), e.g. diesel fuel MS has larger contribution of n-alkanes whereas the lubricant oil MS is enriched in cycloalkanes and aromatics (Tobias et al. 2001). It has been suggested that lubricant oil dominates fuel as a source of primary organic aerosol under typical operating conditions of an engine (Tobias et al., 2001; Worton et al., 2014; Dallmann et al., 2014). However, separating organic species emitted from diesel and gasoline vehicles has proved to be difficult. Vaying levels of diesel trucks in vehicle fleet did not result in clear differences in the MS of organics measured with the SP-AMS (Dallmann et al., 2014).



The composition of organics was studied more carefully with the data collected at Pitkäjärvi. Pitkäjärvi was selected for the detailed investigation as the ratio of organics to BC changed with distance only at Pitkäjärvi suggesting that the traffic particles underwent some atmospheric processing during the dilution. The average MS for organics measured at the highway, roadside, gradient and background at Pitkäjärvi are shown in Fig. S6a and the corresponding MS after the subtraction of

background in Fig. 6b. As already discussed, the MSs at the highway, roadside and gradient were dominated by hydrocarbon fragments and after the background subtraction hydrocarbon fragments from roadside and highway fell into straight line (Fig. S6b). The ratios of the hydrocarbon fragments were slightly different between gradient and roadside (Fig. S6c). Organics had more $C_3H_5^+$ fragment at $m/z$ $41$ during the gradient than at the roadside. Regarding the oxidized fragments, most of them were larger at the background than at the highway, roadside and gradient, shown by negative values in Fig. 6b, except $CO^+$ and

$CO_2^+$ that were clearly higher near the road. Most distinctive negative oxidized fragment was $C_2H_3O^+$ at $m/z$ $43$, especially in the MS measured at the highway and roadside. When studying the behavior of $C_2H_3O^+$ with the distance more closely, it was found that its concentration was smaller only at the highway and roadside but after that it increased to a steady level and remained there for the rest of the gradient.

It was evident that the increase in organics relative to BC (Fig. 5a) was caused by the increase of hydrocarbons via

condensation. By summing all hydrocarbon and oxidized fragments in the MS, the fraction of hydrocarbon fragments in organics increased when the distance to the road got larger, especially after 150 meters from the road (Fig. 5b). In line with that the fraction of oxygen-containing organic fragments decreased with the distance, the descent being more pronounced for the organic fragments with one oxygen atom.

### 3.3.2 Mass size distributions

The evolution of particle chemistry during the dilution was also seen in the mass size distributions of chemical components. At Pitkäjärvi, the size distributions were measured only at the roadside and background (Fig. S7) but it is clearly observable that a mode found at the roadside at ~100 nm, disappeared almost totally when the size distributions were measured at the background location. This mode was dominated by organics (hydrocarbons) and rBC, similar to the previous studies measured by the AMS in traffic environments (e.g. Schneider et al., 2008; Canagaratna et al., 2010; Massoli et al., 2012; Lee et al.,

2015). The mode at ~100 nm was found to be less volatile than the particles in the smaller and larger mode, measured by the two ELPIs and TD (Fig. 4), indicating that the material not evaporated in the TD was the rBC core of the particles. A decrease of the mass in the TD in this mode was likely due to hydrocarbon species. The second mode (at ~300-400 nm) was observed both at the roadside and at the background (Fig. S7). At the background, this mode was mostly made of oxygen-containing organic fragments and sulfate, whereas at the roadside there was also some rBC present. The composition of the second mode

was very similar to that found at Massoli et al. (2012) for the particles upwind of Long Island Expressway. Based on the results from the SP-AMS in the laser-only configuration, they observed a mode at ~500 nm for rBC that was heavily coated with organic material. However, they also suggested that the majority of the mode peaking at ~500 nm consisted of organics and sulfate, that was not associated with rBC cores.



Similar to Pitkäjärvi, also at Herttoniemi there was a mode at ~100 nm (Fig. 7). In contrast to the number size distributions (Fig. 3) the mass size distributions at Herttoniemi changed with the distance from the road. For rBC, the peak of the mode was at 104 nm at the roadside whereas at 30 meters from the road the mode had become narrower and the maximum of the mode had moved to 113 nm. At 40 meters from the road the peak of the rBC mode was found already at 125 nm, and at the distance of 50 m it was at 148 nm. In this study, the dominant mode for rBC was at a significantly smaller size than the mode obtained earlier for EC at the same site at Herttoniemi, 65 meters from the road (Saarikoski et al., 2008). In the earlier study, the maximum of the EC mode was between 300–500 nm. However, the used measurement technique was quite different from the SP-AMS as the EC size distribution was measured by using a small deposit area low pressure impactor with quartz substrates that were analyzed in the laboratory with a thermal-optical transmittance method. Besides the measurement technique, also the meteorology as well as traffic volume could have been quite different.

The shift of the ~100 nm mode is plausible due to the condensation of hydrocarbons on the rBC particles in dilution. When studying the size distribution of hydrocarbons, it was observed that at the roadside and at 30 meters from the road the maxima for hydrocarbons and rBC were at nearly similar sizes but after that, especially at 40 meters from the road, hydrocarbons peaked at the smaller size than rBC (Fig. S8). Also the relative concentration of hydrocarbons was larger than that of rBC at 40 meters. This finding suggests that hydrocarbons were in the same particles with the rBC, and most likely condensed on the surface of the rBC particles at 40 meters. However, at 50 meter distance, the ratio of hydrocarbons to rBC was again similar to that at the roadside and 30 meters from the road. The behavior at a 50 m distance is difficult to explain, however, emissions from vehicles at the nearby street and parking lot probably disturbed our gradient measurements. The size distributions of rBC were obtained from the UMR pToF-data of the SP-AMS by using $m/z$ 36 as a surrogate for rBC as $C_3^+$ at $m/z$ 36 was the strongest carbon cluster signal in the rBC MS. Similarly, $m/z$ 57 in UMR PToF-data was used as a surrogate for hydrocarbons. The size distribution traces $m/z$ 36 and 57 were normalized to the mass concentrations of the corresponding species (rBC, $C_XH_Y^+$) obtained from the high-resolution analysis.

### 3.3.3 Metals

The laser vaporizer used in the SP-AMS extends the range of chemical species detected by the AMS to include refractory species associated with rBC containing particles, such as metals and other elements (Onasch et al., 2012). Standard AMS has a tungsten vaporizer that is heated only up to 600 °C that is not enough for the fast vaporization of metals, however, some metals have been measured with the regular AMS without the laser (e.g. Salcedo et al., 2010, 2012). In this study, iron (Fe), vanadium (V), zinc (Zn) and aluminum (Al) were detected in the particles. At Herttoniemi, a clear gradient was found for Fe, Al and V whereas for Zn the concentration remained high until 50 meters from the road and dropped suddenly (Fig. 8a). At Pitkäjärvi, the concentrations of metals decreased slower than at Herttoniemi, and for all the metals there was a small increase at 75 meters from the road (Fig. 8b). That "bump" could not be explained. At Malmi and Itä-Pakila the concentration of metals did not change with the distance from the road (Fig. S9). That was probably due to the multiple sources of metals at those sites, which inhibited the observation of the concentration gradients from the highway.



In general, the concentration levels for metals were rather similar at Herttoniemi, Pitkäjärvi and Malmi whereas at Itä-Pakila the concentration of vanadium was significantly elevated. Vanadium can be found in the particles from heavy oil combustion (Carbone et al., 2015) but at Itä-Pakila there was no clear heavy oil combustion source nearby that could explain the elevated concentrations. Therefore vanadium was likely to be long-range transported together with sulfate and nitrate that had elevated concentrations at the same time at Itä-Pakila (Fig. S4). However, vanadium is used in catalysts (e.g. Blum et al., 2003) which might partly explain its concentration gradients in Fig. 8.

Aluminum and iron are were metals which can also originate from materials used at road surfaces, tires and brakes, whereas, zinc, phosphorus and magnesium are usually associated with lubricant oils (e.g. Pirjola et al., 2015; Rönkkö et al., 2014; Sodeman et al., 2006). Dallmann et al. (2014) detected zinc and phosphorus in the exhaust plumes of individual trucks by the SP-AMS. They noticed that the ratio of zinc and phosphorus to organics in the emission plume was consistent with typical weight fractions of additives often used in lubricant oils, and used that as an evidence that a large fraction of organics in gasoline exhaust originates from lubricating oil. In this study, phosphorus or magnesium was not detected in particles. The concentrations of metals were calculated from the signal values (in Hz) given by the SP-AMS by using the relative ionization efficiencies measured in the study of Carbone et al. (2015). Unfortunately, the size distributions were not obtained for the metals as the PToF data was saved only in UMR mode and the contributions of metals to the total signal measured at UMR $m/z$'s were very small.

## 3.4 Emission factors

Fleet emission factors, based on the data while driving on the highways, were calculated for CN, PM$_{2.5}$, BC, organics, NO, NO$_2$ and NO$_x$. The average fleet emission factors for CN (particle diameter > 2.5 nm) were found to range from $4.9 \times 10^{15}$ to $1.2 \times 10^{16}$ # (kg fuel)$^{-1}$ (Table 3). The highest value was observed at Pitkäjärvi where the fraction of HD vehicles was highest (Table 1), and lowest at Herttoniemi where the HD fraction and also the total vehicle count were low. At Itä-Pakila, although the HD fraction was as small as at Herttoniemi, the total vehicle count was 51% greater and concequently, the average EF$_{CN}$ was higher $6.5 \times 10^{15}$ # (kg fuel)$^{-1}$. These results and Fig. S10 show that the fraction of HD vehicles has significant effect on the fleet emissions factor of CN.

In general, these results are slightly lower than the value of $9.3 \times 10^{15}$ # (kg fuel)$^{-1}$) presented by Yli-Tuomi et al. (2004), who also performed measurements on the highways in the Helsinki metropolitan region. Our results are in agreement with the ones reported by Massoli et al. (2012) (mixed fleet: $5.3 \times 10^{15}$ # (kg fuel)$^{-1}$), Westerdahl et al. (2008) (LD: $1.8 \times 10^{15}$ and HD: $11 \times 10^{15}$ # (kg fuel)$^{-1}$) and Ježek et al. (2015) (LD$_{gasoline}$:$1.95 \times 10^{15}$, LD$_{diesel}$:$4.4 \times 10^{15}$ and HD$_{diesel}$ (goods vehicles): $11.5 \times 10^{15}$ # (kg fuel)$^{-1}$).

Contrary to the particle number emission factors, all the mass emission factors, EF$_{PM2.5}$, EF$_{BC}$ and EF$_{Org}$, were lowest at Pitkäjärvi. One should remember that there the nucleation mode was very strong but it has only a small effect on the mass emissions. Additionally, the Aitken and soot mode concentrations were smaller than on the highways at the other sites (Fig. 3).



The EFs of NO found here (Table 3) were lower while the EFs of $NO_2$ were higher than the values of $10\pm19$ g (kg fuel)$^{-1}$ for $EF_{NO}$ and $2\pm5$ g (kg fuel)$^{-1}$ for $EF_{NO2}$ reported by Yli-Tuomi et al. (2004). The increased $EF_{NO2}$, with respect to the results of Yli-Tuomi et al. (2004), could be due to the higher direct $NO_2$ emissions of modern diesel cars as the fraction of light duty diesel vehicles of the passenger cars in Finland rose from 18.6 % to 34.3 % in the period between the measurements in years

5 2003-2015 (Official statistics of Finland, 2015). Carslaw et al. (2013) report that in London, the $NO_x$ emissions reduced only from the gasoline fuelled vehicles over the past 15-20 years although the modern diesel vehicles were equipped with after-treatment systems, including SCR systems, designed to reduce $NO_x$ emissions. Furthermore, the authors report that for the diesel passenger cars the relative amount of $NO_2$ was increased as the $NO_2/NO_x$ ratio was 10-15% for Euro III and older type vehicles whereas it was 25-30% for Euro IV-V type vehicles. Ježek et al. (2015) observed reductions in the $EF_{NOx}$ for passenger

10 cars and diesel heavy goods vehicles but no reduction for diesel passenger cars compared to the ten or more years old ones.

## 4 Summary and conclusions

The traffic emissions downwind from the four highways at the Helsinki metropolitan region were measured from 22$^{th}$ October to 6$^{th}$ November 2012 with the mobile measurement platform Sniffer. Measurements were conducted at four locations, within the traffic at the highway, at the roadside, at several distances from the highway (gradient), and at the background. As the

15 pollutants dispersed away from the road, their concentration decreased mostly due to dilution and mixing with the background air. Concentration gradients were observed for the traffic related pollutants CN, $PM_{2.5}$, BC, organics, NO and $NO_2$, and for some metals. Furthermore, a change in the particle number and volume size distribution was noticed. The flow dynamics in the different environments appeared to be an important factor for the pollution dilution. The open environment of Pitkäjärvi produced smooth pollution gradients on the most runs while more complex urban environments of Herttoniemi and Malmi had

20 considerably more randomness present in the gradients. The noise barrier at Itä-Pakila site might lower the pollutant levels considerably by increasing air mixing. Although the traffic pollutants near the highways seemed to vary greatly depending on meteorological conditions and flow dynamics, the results obtained in this study confirm that people living close to high traffic roads are generally exposed to pollutant concentrations that are even double or triple of those measured at 200 m or more away from the road.

25 Traffic particles in the $PM_1$ size fraction mostly consisted of organics and BC. The contributions of traffic related organics and BC stayed rather similar during dilution of emissions (gradient measurements), however, at the most open site (Pitkäjärvi) the relative concentration of organics to BC increased with the distance to the highway. That additional organic mass seemed to consist mostly of hydrocarbons. No evidence of the oxidation of traffic-related organics was found. It was not a surprise as the oxidation of particles occurs in a much longer time period than few minutes covered in this study. Additionally, the

30 measurements were carried out in autumn when solar radiation and therefore oxidant concentrations were small. Particles also contained some metals. Aluminum, iron and vanadium had concentration gradients at Herttoniemi and Pitkäjärvi suggesting them to originate from traffic. Zinc decreased with a distance from the highway only at Herttoniemi.



Regarding number/volume size distributions, particle growth along the gradient was not observed, the particle growth was only visible when comparing fresh emissions to background conditions. However, the mass size distributions at Herttoniemi, measured with the SP-AMS, showed a visible shift of the mode, detected at ~100 nm at the roadside, to a larger size when the distance to roadside increased. That mode consisted mostly of rBC and hydrocarbons and was found to be relatively low

5   volatile.

The fleet average emission factors for particle numbers appeared to be somewhat lower than those reported by Yli-Tuomi et al. (2004). Conversely, the emission factor for $NO_2$ showed an increase. The likely reason is the increased fraction of LD diesel vehicles over the ten years. The fraction of heavy duty traffic, although constituting less than 10 % of the total traffic flow, was found to have a large impact on the emissions.

*Acknowledgements.* The MMEA project was supported by Tekes (the Finnish Funding Agency for Technology and Innovation) **a**nd coordinated by the Finnish energy and environment cluster - CLEEN Ltd. This research was also partly founded by Academy of Finland (funding no 259016), European Social found (SPIRIT, contract no. P-MR-10/04) and the EUROSTARS grant E!4825 FC Aeth. I. Ježek and G. Močnik are employed in Aerosol d.o.o. where the Aethalometer was developed and is

manufactured. The authors are very grateful to Mr. Aleksi Malinen and Mr. Kaapo Lindholm Metropolia University of Applied Sciences for technical expertise and operation of Sniffer.

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

Table 1. Average of hourly meteorological conditions as recorded by the HSY at the Ämmässuo site, representing the regional air mass properties. The traffic LD and HD data are based on the manual three minute observations, and the annual mean traffic flow per day by the Finnish Transport Agency.



| Site | T °C | RH % | Wind direction ° | Wind speed m/s | Traffic LD veh/h | Traffic HD veh/h | % of HD | Annual mean veh/day |
|------|------|------|------------------|----------------|------------------|------------------|---------|---------------------|
| Pitkäjärvi | 1.8 ± 2.3 | 83 ± 12 | 326 ± 30 | 2.7 ± 1.7 | 3400 | 320 | 8.7 | 40000 |
| Malmi | 3.1 ± 1.9 | 84 ± 5.8 | 155 ± 23 | 4.6 ± 1.2 | 4400 | 230 | 4.8 | 55000 |
| Itä-Pakila | 4.7 ± 2.0 | 89 ± 7.0 | 143 ± 10 | 4.4 ± 0.8 | 5900 | 160 | 2.8 | 57000 |
| Herttoniemi | 0.8 ± 2.8 | 77 ± 13 | 257 ± 30 | 2.4 ± 1.1 | 3900 | 110 | 2.9 | 50000 |





Table 2. Mean pollutant concentrations along with standard deviations (std) at the highway (HW), roadside (RS) and background (BG) at the four sites. The distance from the curb to the exact measurement location is indicated in the brackets following RS. CN was measured by the CPC, BC by the aethalometer and Org, $NO_3$, $SO_4$ and $NH_4$ by the SP-AMS.

| | Herttoniemi | | | Malmi | | | Itä-Pakila | | | Pitkäjärvi | | |
|---|---|---|---|---|---|---|---|---|---|---|---|---|
| | HW | RS (11 m) | BG | HW | RS (11 m) | BG | HW | RS (6 m) | BG | HW | RS (14 m) | BG |
| CN | 7.72 | 4.23 | 0.71 | 8.60 | 10.80 | 1.09 | 8.63 | 5.22 | 1.00 | 12.3 | 10.3 | 0.89 |
| std ($\times 10^4$ cm$^{-3}$) | 9.05 | 4.41 | 0.20 | 9.86 | 7.28 | 0.20 | 9.68 | 4.60 | 0.36 | 13.9 | 6.1 | 0.92 |
| $PM_{2.5}$ | 12.9 | 7.3 | 2.2 | 16 | 14.5 | 10.5 | 19.5 | 15.5 | 9.9 | 10.1 | 8.2 | 5.3 |
| std (µg m$^{-3}$) | 11.0 | 4.0 | 0.5 | 22.5 | 4.9 | 4.5 | 11.6 | 4.6 | 2.5 | 8.9 | 6.3 | 5.8 |
| BC | 6.08 | 4.95 | 0.57 | 6.84 | 4.26 | 0.63 | 4.68 | 2.99 | 0.65 | 4.45 | 3.52 | 0.43 |
| std (µg m$^{-3}$) | 9.29 | 5.99 | 0.27 | 12.8 | 3.23 | 0.30 | 5.34 | 1.93 | 0.31 | 5.02 | 1.96 | 0.30 |
| Org | 4.55 | 3.22 | 1.33 | 7.83 | 6.01 | 3.37 | 9.01 | 4.56 | 3.62 | 5.54 | 3.36 | 0.97 |
| std (µg m$^{-3}$) | 5.72 | 1.38 | 0.43 | 11.3 | 4.61 | 1.15 | 13.6 | 2.80 | 1.00 | 5.07 | 2.88 | 0.73 |
| $NO_3$ | 0.093 | 0.095 | 0.078 | 0.58 | 0.62 | 0.73 | 0.91 | 0.88 | 0.94 | 0.19 | 0.13 | 0.11 |
| std (µg m$^{-3}$) | 0.047 | 0.072 | 0.036 | 0.40 | 0.43 | 0.50 | 0.70 | 0.90 | 0.93 | 0.19 | 0.13 | 0.12 |
| $SO_4$ | 0.36 | 0.29 | 0.18 | 1.3 | 1.1 | 1.4 | 1.5 | 1.0 | 1.2 | 0.57 | 0.38 | 0.53 |
| std (µg m$^{-3}$) | 0.23 | 0.20 | 0.057 | 1.1 | 0.88 | 0.88 | 0.03 | 0.26 | 0.07 | 0.32 | 0.22 | 0.34 |
| $NH_4$ | 0.10 | 0.088 | 0.057 | 0.51 | 0.50 | 0.56 | 0.68 | 0.54 | 0.61 | 0.20 | 0.13 | 0.15 |
| std (µg m$^{-3}$) | 0.075 | 0.076 | 0.003 | 0.40 | 0.38 | 0.38 | 0.21 | 0.33 | 0.27 | 0.12 | 0.008 | 0.10 |
| NO | 132 | 52.4 | 0.4 | 150 | 107 | 2.0 | 94.5 | 65.4 | 3.1 | 138 | 105 | 7.4 |
| std (µg m$^{-3}$) | 146 | 48.6 | 0.8 | 348 | 69.5 | 0.9 | 95.8 | 43.4 | 1.9 | 142 | 57.6 | 32.3 |
| $NO_2$ | 73.0 | 46.0 | 12.3 | 54.2 | 76.6 | 18.0 | 61.4 | 50.7 | 19.2 | 54.3 | 62.1 | 15.3 |
| std (µg m$^{-3}$) | 89.6 | 27.4 | 3.3 | 284 | 39.2 | 5.6 | 57.9 | 30.3 | 7.6 | 75.2 | 33.3 | 41.0 |
| $NO_X$ | 205 | 98.4 | 12.6 | 204 | 184 | 20.0 | 156 | 116 | 22.3 | 192 | 167 | 22.7 |
| std (µg m$^{-3}$) | 190 | 70.5 | 3.30 | 304 | 97 | 5.9 | 125 | 59.0 | 8.8 | 176 | 78.8 | 67.7 |





Table 3. Average emission factors along with standard deviations calculated from the measurements on the highways.

| | | Herttoniemi | Malmi | Itä-Pakila | Pitkäjärvi |
|---|---|---|---|---|---|
| $EF_{CN}$ (# (kg fuel)$^{-1}$) | Mean | $4.9 \times 10^{15}$ | $6.1 \times 10^{15}$ | $6.5 \times 10^{15}$ | $11.6 \times 10^{15}$ |
| | Std | $6.5 \times 10^{15}$ | $5.7 \times 10^{15}$ | $7.1 \times 10^{15}$ | $14.6 \times 10^{15}$ |
| $EF_{PM2.5}$ (g (kg fuel)$^{-1}$) | Mean | 0.69 | 0.63 | 0.85 | 0.32 |
| | Std | 0.17 | 0.16 | 0.10 | 0.38 |
| $EF_{BC}$ (g (kg fuel)$^{-1}$) | Mean | 0.43 | 0.54 | 0.30 | 0.15 |
| | Std | 0.67 | 0.65 | 0.22 | 1.14 |
| $EF_{Org}$ (g (kg fuel)$^{-1}$) | Mean | 0.26 | 0.33 | 0.33 | 0.24 |
| | Std | 0.33 | 0.20 | 0.17 | 0.13 |
| $EF_{NO}$ (g (kg fuel)$^{-1}$) | Mean | 8.12 | 9.86 | 7.44 | 11.48 |
| | Std | 6.31 | 9.65 | 5.28 | 7.60 |
| $EF_{NO2}$ (g (kg fuel)$^{-1}$) | Mean | 4.14 | 4.02 | 3.45 | 4.47 |
| | Std | 4.44 | 7.26 | 3.95 | 5.79 |
| $EF_{NOx}$ (g (kg fuel)$^{-1}$) | Mean | 12.2 | 14.1 | 10.9 | 16.5 |
| | std | 8.0 | 12.2 | 6.8 | 11.8 |



Figure 1. Four gradient measurement locations in the Helsinki region. In the subplots, the black arrows show the driving direction on the gradient roads and BG depicts the local background measurement sites. Also shown is a meteorological measurement site at Ammässuo. (OpenStreetMap)







Figure 2. Normalized particle number ($N_{tot}$) and mass concentration ($PM_{2.5}$), BC and organics concentration, as well as NO and $NO_2$ concentration as a function of distance from the highway at four measurement locations shown in the legend. Zero distance refers to the edge of the road, and negative values to driving on the highway. Also shown are the fitted reduction curves. Background values were subtracted from the measured concentrations.





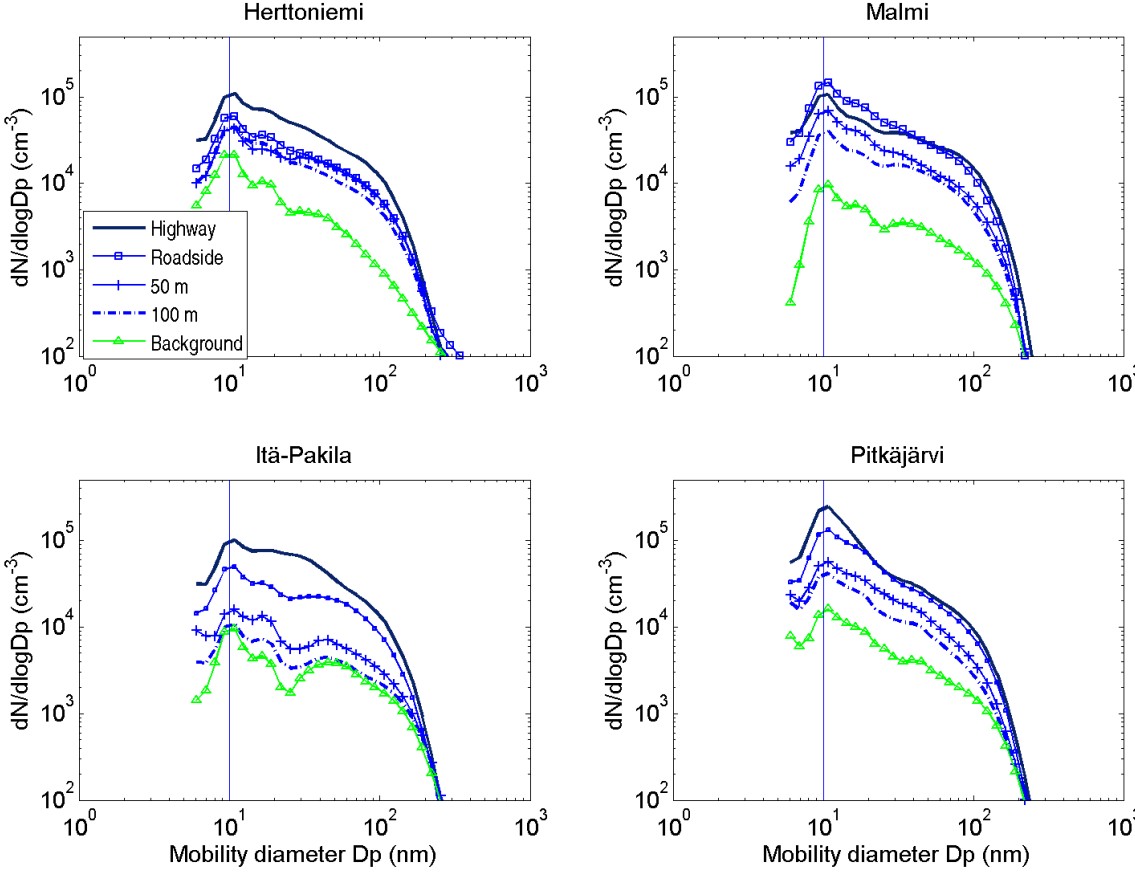

Figure 3. Particle size distribution as measured by the EEPS at different distances from the roadside on the four locations.





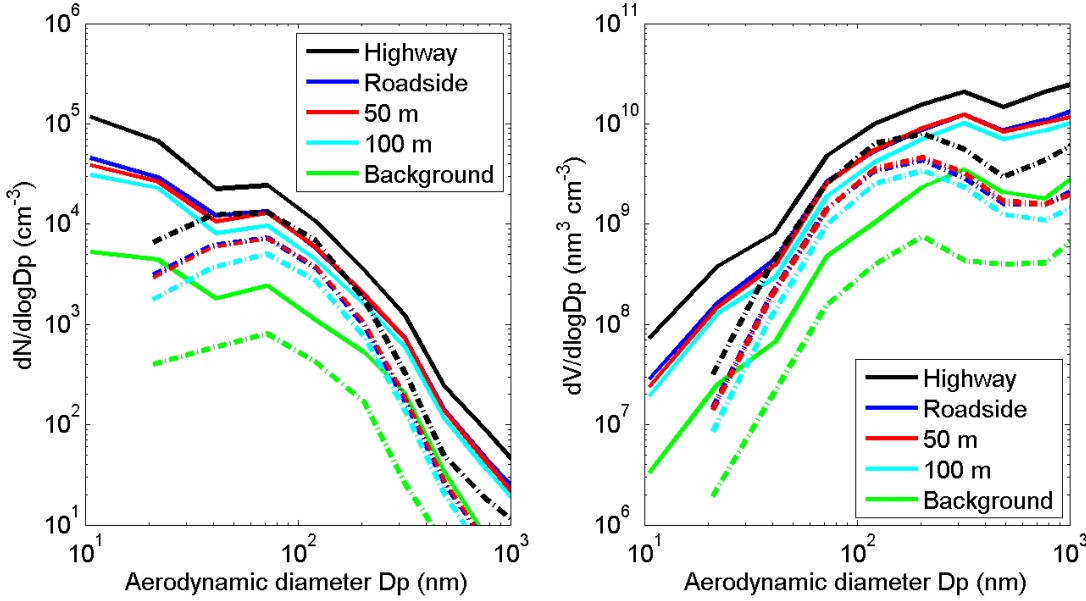

Figure 4. Average particle number size distribution (left) and volume size distribution (right) measured at different distances
from the highway at Herttoniemi with two ELPIs, one measured before (solid lines) and the other after (dash dot lines) the
5    thermodenuder. Note that x-axis refers to aerodynamic diameter.





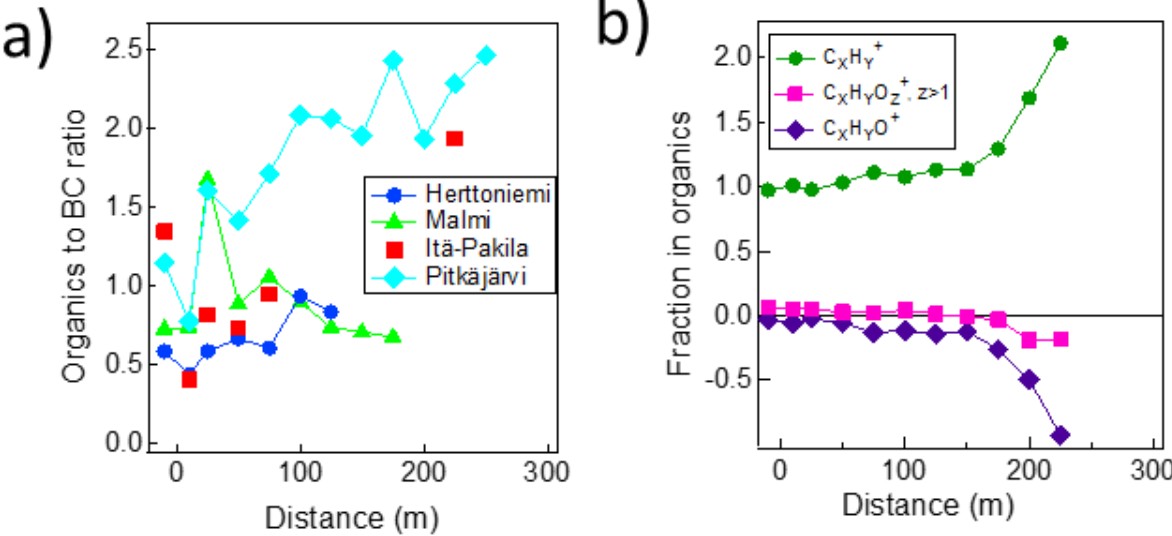

Figure 5. The average ratio of organics to BC at four sites (a) and the fractions of hydrocarbon fragments ($C_XH_Y^+$) and oxidized organic fragments ($C_XH_YO^+$ and $C_XH_YO_Z^+$, $_{z>1}$) at Pitkäjärvi (b) as a function of the distance from the highway. Zero distance refers to the roadside and negative value driving on the highway. Background values were subtracted from the measured concentrations before calculating the ratios.




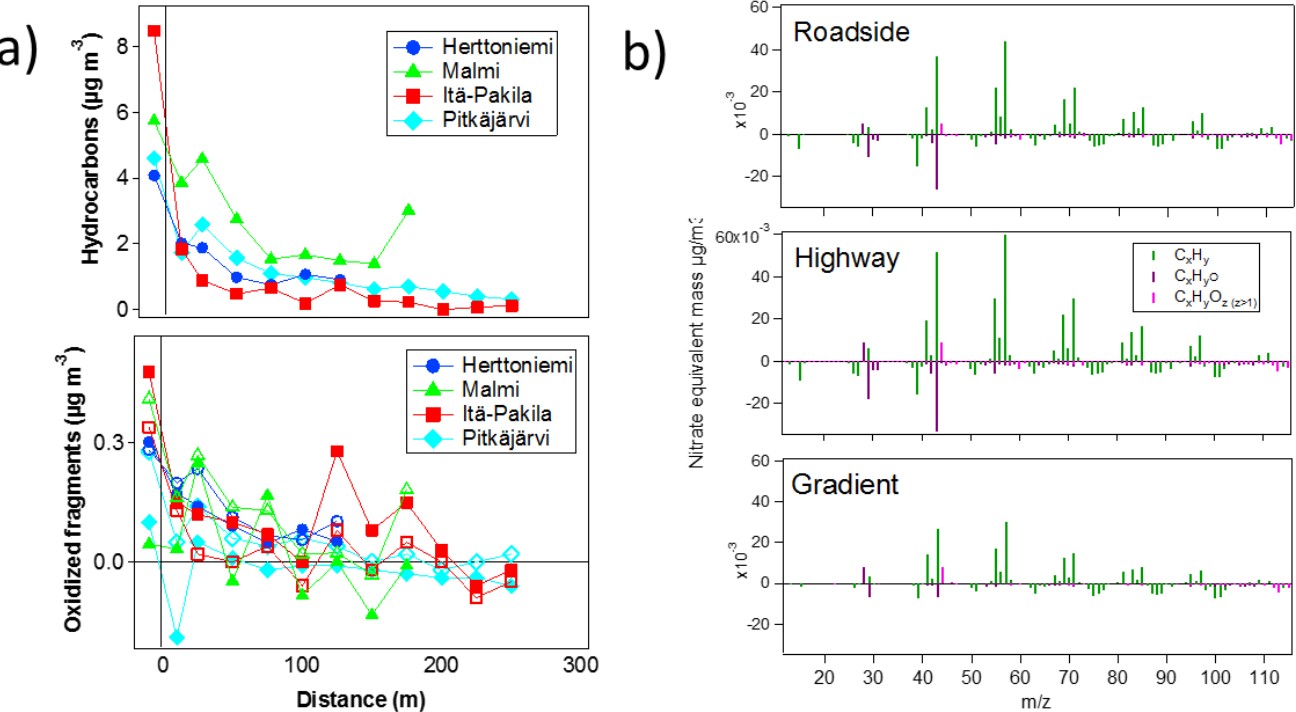

Figure 6. Concentrations of hydrocarbon fragments ($C_XH_Y^+$) and oxidized organic fragments ($C_XH_YO^+$ and $C_XH_YO_Z^+$, $_{Z>1}$)
measured at the four sites (a), and the average mass spectra for highway, roadside and gradient at Pitkäjärvi (b). Solid markers
refer to organic fragments with one oxygen atom and open markers to organic fragments with more than one oxygen atoms in
lower figure of (a). In (a) zero distance refers to the roadside and negative value driving on the highway. Background values
were subtracted both from the measured concentrations and mass spectra.





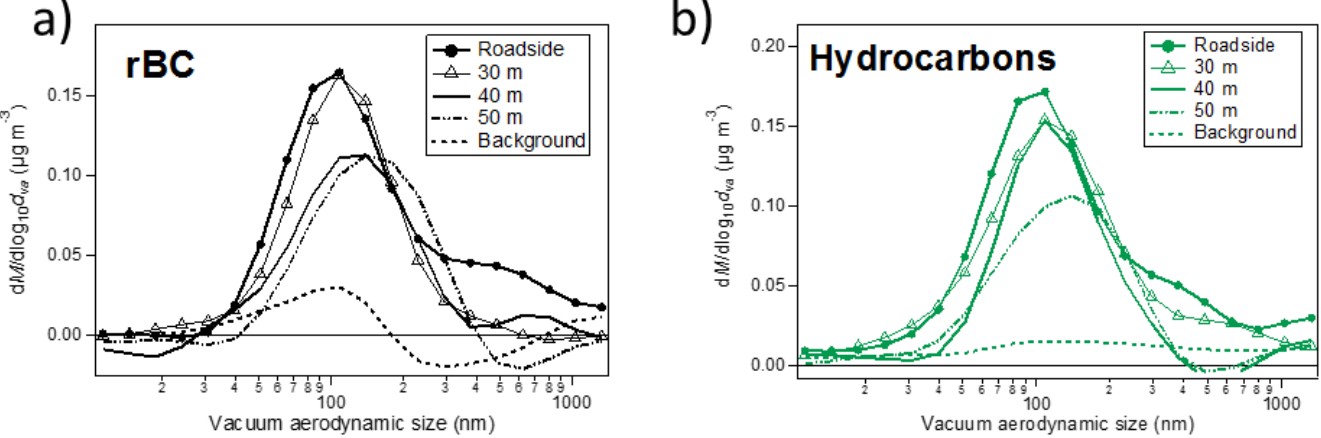

Figure 7. Particle mass size distribution at Herttoniemi on 26th of Oct 2012. *m/z 36* was used as a surrogate for rBC and *m/z 57* for for hydrocarbons.




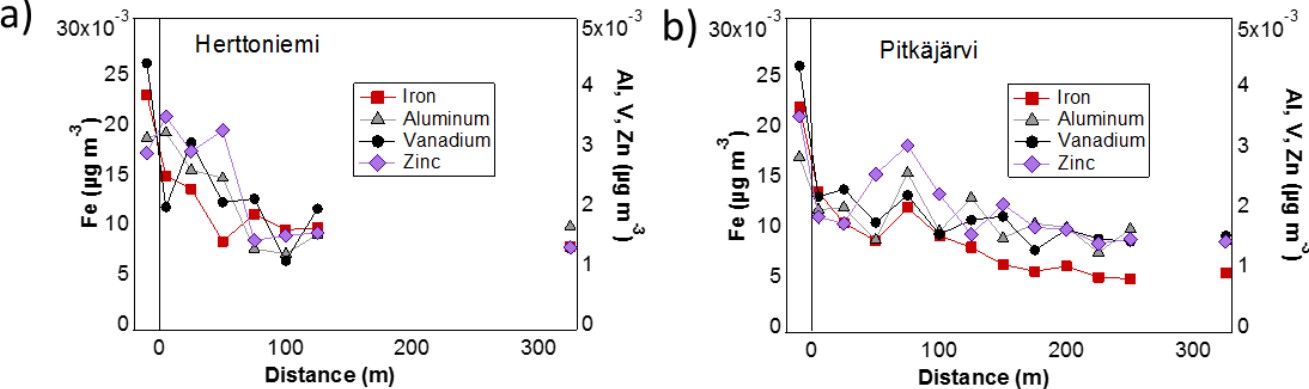

Figure 8. Concentrations of iron, aluminum, vanadium and zinc at Herttoniemi (a) and Pitkäjärvi (b) as a function of the distance from the highway. Zero distance refers to the roadside, negative values to driving on the highway and right-most point to the background location. Background values are not subtracted from the measured concentrations. Gradients for Itä-Pakila and Malmi are presented in supplement.