# Peer review of "Chemical and physical characterization of traffic particles in"

_Atmospheric Chemistry and Physics, 2015_

## Referee Comment (RC1) · Anonymous Referee #1 · 19 Feb 2016

General comments

The paper by Enroth et al., titled "Chemical and physical characterization of traffic particles in four different highway environments in the Helsinki metropolitan area" addresses a relevant topic for air quality. Traffic-related pollution in urban centers is likely the largest sources of harmful emissions especially at the global level. Even in North America and Europe, where the air quality has improved immensely in the last decades due to stringent emissions controls and overall more efficient, cleaner vehicles, particle and gas pollutants from traffic still represent an health hazard.

This paper present a comprehensive summary of gas phase and particle measurements of roadside pollution made with state of the art techniques using a mobile laboratory facility. Despite the topic and the technique used for the study are not new, the paper still report useful data and it does address relevant scientific questions within the scope of ACP. The study shows pollution gradients at four different sites within the Helsinki metropolitan area, and both particle and gases are reported. The particles are speciated with an high resolution mass spectrometer (the Aerodyne Research SP-AMS) that has the additional capability of measuring black carbon via laser vaporization Both mass concentrations and particle size distributions are reported.

The most novel aspect of this paper, compared to other similar studies, is the section reporting measurements of metals associated with traffic emissions. In addition, the results regarding the increased fleet average emission factor for NO2 are worth noting, because, as the authors suggest, they are very much a consequence of the increase of diesel cars in Europe over the last decade.

The paper is well written and the results are presented in a straightforward manner, with an appropriate amount of references to previous work, figures and supplementary material. The experimental methods are explained clearly and the amount of data is sufficient to support the statements and conclusions of the paper. I do not see any flaw with the data or the methods used, therefore I recommend publication after minor technical corrections, listed below. English and punctuation should be checked through the paper

Specific comments

Page 2, line 9: remove year before 2012 Page 2, line 15: rewrite part of the phrase "....to background levels 300-500 m downwind from the roadside". Page 3, line 29: remove 'were" Page 8, line 12: rewrite as: "Similarly, Massoli et al. (2012) did not....."

Figures

Figure S6: I suggest to add the mass spectrum of BC as well (at least from C1 to C5)

---

## Referee Comment (RC2) · Anonymous Referee #2 · 7 Mar 2016

General comments This paper presents the finding of a multi week field study performed in Helsinki, Norway that focussed on emissions from traffic and how those emissions impacted nearby ambient air and changed with distance from the traffic sources. Much of the data were collected with an instrumented mobile van that slowly transited from the road side of several roads. High time resolution instrumentation was included to allow one second collection of particle size composition, particle numbers, and common traffic-generated gaseous and particle phase chemistry. Changes in particle size distribution and composition were also assessed with distance from the roadway. Finally, emission factors were developed for common traffic related pollutants. The information presented makes a very important contribution to the complex issues

of the dynamic processes that occur once pollutants are emitted and moves from the roadway to the community. It is quite comprehensive and only a few technical question appeared as it was reviewed. Upon consideration of these comments and others from the reviewing community the paper should prove quite useful.

Specific comments Page 6, line 5—the weather was described as "rather mild". The conditions include data collection under what most researchers might consider quite cold and wet. It appears that sampling was conducted in sub-freezing at one site and most all was performed when temperatures were below 5 degrees C. Average humilities were approximately 90% at one site. The terminology of "mild" is important for clarification, however, much more important is that the nature of the study was to consider the dynamic processes that occur between tailpipe and the first few hundred meters from a roadway. It is well known that temperature plays a key role along with concentration in these processes. It is likely that humidity is also important. It clearly critical for PM related factors, but could even play a role in NOx conversions observations. Thus one very key point for further including in the paper is a science-based appraisal of the somewhat extreme conditions should be viewed, how they might impact particle and gas dynamics how others might use the data collected and conclusions drawn. Page 6, line 28—PM 2.5 mass data were produced by a DustTrak. The operational conditions of this unit were not described beyond inclusion of a mass calibration factor from a prior study published in 2012. The calibration factor reported in that study was 1.46. This is a critical correction factor and the findings related to PM 2.5 would be far more supportable had a proper contemporary calibration factor been made. Further, there is no mention of whether humidity was considered as the data were used while average humilities at one site were 89%. The authors are suggested to raise this point for caution to reader and if possible, should address what was done and perhaps quickly determine a mass calibration for the instrument to either confirm it is appropriate, to correct the data or perhaps consider elimination of PM mass data entirely. It is not a key factor in this study. Some data from nearby ambient monitoring stations may also be used to evaluate the correction factor in the paper. Page 17, line 28—related

to the PM2.5 points above—should either the calibration or Rh considerations prove troublesome to correct It is at least important to inform the reader of possible problems with this data.

---

## Author Comment (AC1) · 7 Apr 2016

**Reply to Referee #1**

We thank Referee #1 for positive evaluation of the manuscript. All suggested technical corrections have been made in the revised manuscript. Also English and punctuations have been checked through the paper. The changes made in the manuscript are below written by blue color.

**General comments**
**The paper by Enroth et al., titled "Chemical and physical characterization of traffic particles in four different highway environments in the Helsinki metropolitan area" addresses a relevant topic for air quality. Traffic-related pollution in urban centers is likely the largest sources of harmful emissions especially at the global level. Even in North America and Europe, where the air quality has improved immensely in the last decades due to stringent emissions controls and overall more efficient, cleaner vehicles, particle and gas pollutants from traffic still represent an health hazard. This paper present a comprehensive summary of gas phase and particle measurements of roadside pollution made with state of the art techniques using a mobile laboratory facility. Despite the topic and the technique used for the study are not new, the**
**paper still report useful data and it does address relevant scientific questions within the scope of ACP. The study shows pollution gradients at four different sites within the Helsinki metropolitan area, and both particle and gases are reported. The particles are speciated with an high resolution mass spectrometer (the Aerodyne Research SPAMS) that has the additional capability of measuring black carbon via laser vaporization Both mass concentrations and particle size distributions are reported.**
**The most novel aspect of this paper, compared to other similar studies, is the section reporting measurements of metals associated with traffic emissions. In addition, the results regarding the increased fleet average emission factor for NO2 are worth noting, because, as the authors suggest, they are very much a consequence of the increase of diesel cars in Europe over the last decade.**
**The paper is well written and the results are presented in a straightforward manner, with an appropriate amount of references to previous work, figures and supplementary material. The experimental methods are explained clearly and the amount of data is sufficient to support the statements and conclusions of the paper. I do not see any flaw with the data or the methods used, therefore I recommend publication after minor technical corrections, listed below. English and punctuation should be checked through the paper.**

**Specific comments**

**Page 2, line 9: remove year before 2012**

Done

**Page 2, line 15: rewrite part of the phrase**
**"....to background levels 300-500 m downwind from the roadside".**

The sentence was rephrased: "Generally, all of these studies showed that the pollutant concentrations were higher near highway than further from the roadside, sharply decreasing to background levels within 300-500 m downwind."

**Page 3, line 29: remove 'were"**

This part of the sentence was rephrased "and (4) background measurements for each environment at a suitable remote location approximately 500 m away from the highway."

**Page 8, line 12: rewrite as: "Similarly, Massoli et al. (2012) did not....."**

Done

**Figures**
**Figure S6: I suggest to add the mass spectrum of BC as well (at least from C1 to C5)**

As suggested, the mass spectrum of rBC (fromC1 to C7) was added in Fig. S6a where gradient refers to the average concentration over all distances. Note different scales of the y-axes.

a)

[Figure]

---

## Author Comment (AC2) · 7 Apr 2016

**Reply to Referee #2**

We thank Referee #2 for positive evaluation of the manuscript. All suggested corrections have been made in the revised manuscript. Also English and punctuations have been checked through the paper. The changes made in the manuscript are below written by blue color.

**General comments**
**This paper presents the finding of a multi week field study performed in Helsinki, Norway that focussed on emissions from traffic and how those emissions impacted nearby ambient air and changed with distance from the traffic sources. Much of the data were collected with an instrumented mobile van that slowly transited from the road side of several roads. High time resolution instrumentation was included to allow one second collection of particle size composition, particle numbers, and common traffic-generated gaseous and particle phase chemistry. Changes in particle size distribution and composition were also assessed with distance from the roadway. Finally, emission factors were developed for common traffic related pollutants. The information presented makes a very important contribution to the complex issues of the dynamic processes that occur once pollutants are emitted and moves from the roadway to the community. It is quite comprehensive and only a few technical question appeared as it was reviewed. Upon consideration of these comments and others from the reviewing community the paper should prove quite useful.**

**Specific comments**
**Page 6, line 5ăĂˇ Tthe weather was described as "rather mild". The conditions include data collection under what most researchers might consider quite cold and wet. It appears that sampling was conducted in sub-freezing at one site and most all was performed when temperatures were below 5 degrees C. Average humilities were approximately 90% at one site. The terminology of "mild" is important for clarification, however, much more important is that the nature of the study was to consider the dynamic processes that occur between tailpipe and the first few hundred meters from a roadway. It is well known that temperature plays a key role along with concentration in these processes. It is likely that humidity is also important. It clearly critical for PM related factors, but could even play a role in NOx conversions observations. Thus one very key point for further including in the paper is a science-based appraisal of the somewhat extreme conditions should be viewed, how they might impact particle and gas dynamics how others might use the data collected and conclusions drawn.**

The referee is right, "rather mild" was not the best phrase, even though this kind of weather is rather typical, but not yet extreme, for a few months in Finland and in the northern hemisphere. The sentences (p. 7, lines 21-24) were substituted by: "Typical to Finnish autumn weather, the temperature was around 0.8-4.7$^{o}$C, relative humidity 77-89%, and wind speed around 3-5 m s$^{-1}$, monitored at the meteorological measurement site at Ämmässuo (Fig. 1) by the HSY. The measurement altitude was 15 m so these values represent regional air mass properties."

We added more discussion in Introduction (p. 3 lines 20 - p. 4. line 6) concerning dilution and aerosol dynamics, their dependence on temperature, relative humidity and wind speed, and their effects on particle size distribution during dispersion. "These studies showed that the concentration levels and gradient shapes of UFP and other primary vehicular emissions near major roads depend in a complex way on many factors, including meteorological conditions such as atmospheric stability, temperature, wind speed, wind direction, and surface boundary layer height (Durant et al., 2010). Dilution is a very crucial process, additionally it is accompanied by aerosol dynamics processes such as nucleation, coagulation, condensation, evaporation and deposition (Kumar et al., 2011 and references therein). In the diluting and cooling exhaust new particles are formed by homogeneous nucleation during first milliseconds (Kittelson, 1998), after that they immediately grow by condensation of condensable vapours. Low temperature favours nucleation and condensation, whereas evaporation becomes important during high ambient temperature. On the other hand, the majority of volatile organic compounds is emitted by vehicles during cold starts (Weilenmann et al., 2009). Consequently, Padro-Martinez et al. (2012) report that the gradient concentrations were much higher in winter than in summer, even 2-3 times higher as observed by Pirjola et al. (2006). Also relative humidity might affect PM emissions by vehicles. Typically, in the street scale (around 200 m from the roadside) coagulation is too slow to modify particle size distribution. However, under inefficient dispersion conditions (wind speed < 1m s$^{-1}$) self- and inter-modal coagulation as well as condensation and evaporation might become important transforming the particle size distribution (Karl et al., 2016 and references therein).
Besides dilution and aerosol dynamics, traffic fleet and flow rate (e.g. Zhu et al., 2009; Beckerman et al., 2008), background concentrations (Hagler et al., 2009), and atmospheric chemical and physical processes (Beckerman et al., 2008; Clements et al., 2009), all affect pollutant concentrations near the highways."

As in the referred published studies, also in our study, the results are strongly dependent on the local environmental conditions. Therefore, it is true that readers should always take the environmental conditions into account if the data and drawn conclusions are used in other studies. In Conclusions, we changed the sentence (p.

22, lines 10-15) "Although the traffic pollutants near the highways seemed to vary greatly depending on meteorological conditions and flow dynamics, the results obtained in this study confirm that people living close to high traffic roads are generally exposed to pollutant concentrations that are even double or triple of those measured at 200 m or more away from the road" to "Although the traffic pollutants near the highways seemed to vary greatly depending on meteorological conditions and flow dynamics, the results obtained in this study under these environmental conditions confirm that people living close to high traffic roads are generally exposed to pollutant concentrations that are even double or triple of those measured at 200 m or more away from the road. "

It is well known that for NO to $NO_2$ conversion photochemical production of $O_3$ is important. However, during our campaign solar radiation was weak due to the short sunshine time (7:30 - 16:30) and large zenith angle, besides the measurements occurred during sunrise (7-10 am) and sunset (3-6 pm), and furthermore, most of the days were partly cloudy. This was mentioned on p. 12 "The sunrise and sunset during the measurement period coincided with the rush hours, thus making the analysis of photochemistry more difficult." Photochemical oxidation might not be the reason for high and rather constant normalized $NO_2$ gradients in Malmi (Fig. 2). As explained on p. 12, high number of diesel passenger cars directly emitting $NO_2$ affect the concentrations. It should also be noted that the highway and gradient concentrations were not simultaneous measurements.

Added references:

Canagaratna, M. R., Onasch, T. B., Wood, E.C., Herndon, S.C., Jayne, J. T., Cross, E.S., Miake-Lye, R.C., Kolb, C.E., and Worsnop, D. R.: Evolution of vehicle exhaust particles in the atmosphere, J. Air & Waste Manage. Assoc., 60, 1192-1203, 2010.

Karl, M., Kukkonen, J., Keuken, M.P., Lützenkirchen, S., Pirjola, L., and Hussein, T.: Modelling and Measurements of Urban Aerosol Processes on the Neighbourhood Scale in Rotterdam, Oslo and Helsinki, Atmos. Chem. Phys. Discuss., 15, 35157–35200, 2015

Kittelson, D.B.: Engines and nano-particles: a review, J. Aerosol Sci., 29, 575-588, 1998.

Pirjola, L., Lähde, T., Niemi, J.V., Kousa, A., Rönkkö, T., Karjalainen, P., Keskinen, J., Frey, A., Hillamo, R.: Spatial and temporal characterization of traffic emission in urban microenvironments with a mobile laboratory, Atmos. Environ., 63, 156-167, 2012.

Weilenmann, M., Favez, J.-Y., and Alvarez, R.: Cold-start emissions of modern passenger cars at different low ambient temperatures and their evolution over vehicle legislation categories, Atmos. Environ., 43, 2419-2429, 2009.

**Page 6, line 28ă˘ĂˇTPM 2.5 m might contribute to the evolution ass data were produced by a DustTrak. The operational conditions of this unit were not described beyond inclusion of a mass calibration factor from a prior study published in 2012. The calibration factor reported in that study was 1.46. This is a critical correction factor and the findings related to PM 2.5 would be far more supportable had a proper contemporary calibration factor been made. Further, there is no mention of whether humidity was considered as the data were used while average humilities at one site were 89%. The authors are suggested to raise this point for caution to reader and if possible, should address what was done and perhaps quickly determine a mass calibration for the instrument to either confirm it is appropriate, to correct the data or perhaps consider elimination of PM mass data entirely. It is not a key factor in this study. Some data from nearby ambient monitoring stations may also be used to evaluate the correction factor in the paper.**

Since the DustTrak was not calibrated to the aerosol measured in this study, we decided to eliminate the $PM_{2.5}$ as suggested by the referee. Instead we estimated the $PM_1$ concentration as the sum of the concentrations of the BC measured with the AE33, and the organic and inorganic species measured with the SP-AMS. The $PM_1$ concentrations ($\mu g/m^3$) were added in Fig. 2 and Table 2, and the emission factors $EF_{PM1}$ (g/kg fuel) in Table 3.

In section 2.2 Instrumentation (p. 9, lines 22-23) we added: "In this study, the $PM_1$ concentration was estimated as the sum of the concentrations of BC, measured with the Aethalometer, and the organics and inorganics, measured with the SP-AMS."

**Page 17, line 28ă˘Ă ˘Trelated to the PM2.5 points aboveă˘Ă˘ Tshould either the calibration or Rh considerations prove troublesome to correct It is at least important to inform the reader of possible problems with this data.**

This comment is not relevant any more, since the $PM_{2.5}$ data was eliminated.

---

## Author Response (AR2)

**Response to Editor's comment:**

One aspect that should be clarified is how you quantified the concentration of metals using the SP-AMS in IR laser mode (Fig. 8). This would require some sort of calibration. Did you calibrate the SP-AMS for metals, or are you applying the calibration from a previous paper on the SP-AMS? This important detail will not be difficult to clarify in the text.

We added the following sentences in Section 2.2. Instrumentation, Section 3.3.3 Metals, and in the caption of Fig. 8:

p. 9 lines 6-7: "Default relative ionization efficiencies (RIE) were used for organics, inorganic species and rBC. RIEs for metals were taken from Carbone et al. (2015)."

p. 19, lines 16-21: "In this study, both laser and tungsten vaporizers were installed in the SP-AMS. The concentrations of metals were calculated from the signal values (in Hz) given by the SP-AMS by using the relative ionization efficiencies measured in the study of Carbone et al. (2015). Unfortunately, the size distributions were not obtained for the metals as the PToF data was saved only in UMR mode and the contributions of metals to the total signal measured at UMR *m/z's* were very small." The last two sentences were removed at the end of Section 3.3.3.

Fig. 8 caption: "The concentrations of metals were calculated from the signal values (in Hz) given by the SP-AMS by using the relative ionization efficiencies measured in the study of Carbone et al. (2015)."